# The Role of Glycoside Hydrolases in Phytopathogenic Fungi and Oomycetes Virulence

**DOI:** 10.3390/ijms22179359

**Published:** 2021-08-28

**Authors:** Vahideh Rafiei, Heriberto Vélëz, Georgios Tzelepis

**Affiliations:** Department of Forest Mycology and Plant Pathology, Swedish University of Agricultural Sciences, Uppsala Biocenter, Box 7026, SE-750 07 Uppsala, Sweden; vahideh.rafiei@slu.se (V.R.); heriberto.velez@slu.se (H.V.)

**Keywords:** carbohydrate-active enzymes, cell wall degrading enzymes, pathogenicity, phytopathogens, plant–microbe interactions

## Abstract

Phytopathogenic fungi need to secrete different hydrolytic enzymes to break down complex polysaccharides in the plant cell wall in order to enter the host and develop the disease. Fungi produce various types of cell wall degrading enzymes (CWDEs) during infection. Most of the characterized CWDEs belong to glycoside hydrolases (GHs). These enzymes hydrolyze glycosidic bonds and have been identified in many fungal species sequenced to date. Many studies have shown that CWDEs belong to several GH families and play significant roles in the invasion and pathogenicity of fungi and oomycetes during infection on the plant host, but their mode of function in virulence is not yet fully understood. Moreover, some of the CWDEs that belong to different GH families act as pathogen-associated molecular patterns (PAMPs), which trigger plant immune responses. In this review, we summarize the most important GHs that have been described in eukaryotic phytopathogens and are involved in the establishment of a successful infection.

## 1. Introduction

Plants have evolved effective mechanisms of resistance to cope with pathogen attack. A primary challenge for a pathogen is to breach the host cell wall, which is the fundamental physical barrier protecting plants against microbial attack [1,2]. The plant cell wall is composed of polysaccharides such as cellulose, hemicellulose, and pectin (Figure 1) [3]. To penetrate and break down this barrier, most phytopathogenic fungi and oomycetes have developed an arsenal of tools such as secreting cell wall-degrading enzymes (CWDEs) that include pectinases, polygalacturonases, glucanases, cellulases, and xyloglucanases, to degrade the components of the host cell wall [4]. Enzymes that are responsible for breaking down complex carbohydrates and polysaccharides into smaller products are called carbohydrate-active enzymes (CAZymes) [5].

Most phytopathogenic fungi and oomycetes secrete various kinds of CAZymes, and CWDEs are the most abundant and important enzymes due to their roles in penetration, invasion, and pathogenicity. Although the exact role of the majority of CWDEs have remained unknown, various studies have reported that these enzymes are important virulence factors in many plant pathogens, which can facilitate pathogen invasion and disease development, as well as providing pathogens with nutrition by carbohydrates released from the cell wall [6,7]. On the other hand, pathogen attacks can be recognized by plants through the detection of elicitor molecules, such as hydrolyzed plant cell wall components, which result in the activation of plant immune responses [8].

In general, the CAZymes have been classified into six main classes that include glycoside hydrolases, polysaccharide lyases, glycosyltransferases, carbohydrate esterases, auxiliary activity enzymes, and carbohydrate-binding modules [9]. Comparative analysis of CAZymes across all fungal species indicated that there is a great diversity in the number and types of CAZymes among fungi. For instance, phytopathogenic fungi have the greatest number of CAZymes, with necrotrophic and hemi-biotrophic fungal species having more CAZymes compared to biotrophic ones [10]. In this review, we have summarized the latest knowledge of CWDEs belonging to different Glycoside hydrolases families that are involved in the pathogenicity of fungi and oomycetes.

**Figure 1 ijms-22-09359-f001:**
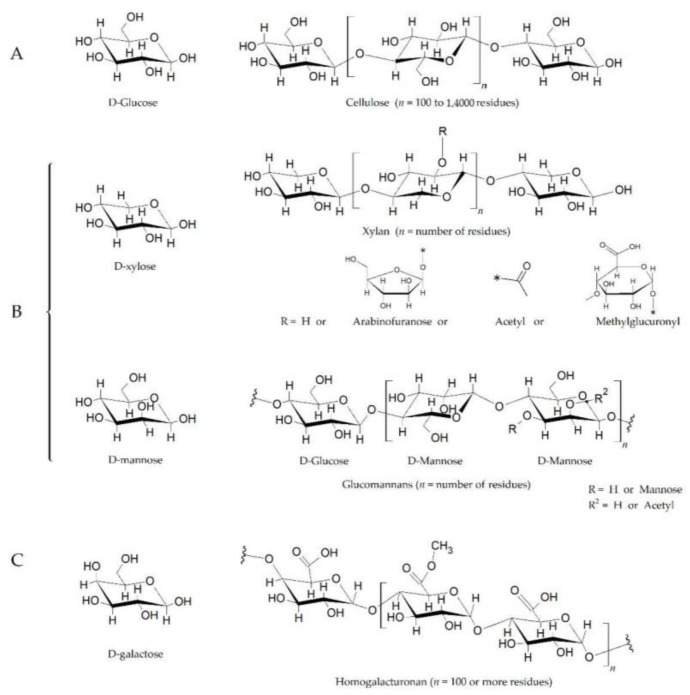
The plant cell wall is composed of polysaccharides such as cellulose, hemicellulose, and pectin. (**A**). Cellulose is a polymer of d-glucose units linked by β1-4 bonds, also known as glucosidic bonds. (**B**). Xylans and glucomannans are two components of hemicellulose. The xylan backbone, which is composed of d-xylose linked together by β1-4 bonds, also contains varying amounts of acetyl, methylglucuronyl, and arabinofuranosyl side chains. The arabinofuranosyl side chains can also be linked to aromatic acids (e.g., p-coumaric acid). Glucomannans are composed of d-glucose and D-mannose at a ratio of 1.6:1 (linked by β1-4 bonds), and with branches of d-mannose residues at the C3 position (β1-3 bonds), on both glucose and mannose. (**C**). Pectin is the methylated ester of polygalacturonic acid, consisting of chains of 300 to 1000 galacturonic acid units joined with 1α→4 bonds. The structure shows three methyl ester forms (−COOCH3) for every two carboxyl groups (−COOH), having a 60% degree of esterification. The substituted residues at C-4 with neutral and acidic oligosaccharide side chain composing of arbinose, galactose, fructose, and glucuronic acid. The length of the chains and the attached residues and their placement varies among plant species [11,12]. The chemical structures have been drawn with ACD/ChemSketch (Freeware Software, version 2020.2.1, www.acdlabs.com, accessed on 20 August 2021).

## 2. Glycoside Hydrolases

Glycoside hydrolases (GHs) are the most predominant and widespread class among the fungi analyzed so far [10]. They are the largest and the most diverse characterized group of protein families in the CAZy database that are responsible for hydrolytic cleavage of the glycosidic bonds between carbohydrate molecules or between a carbohydrate and a non-carbohydrate group (e.g., lipid or sugar) [13,14,15]. GHs consist of 171 protein families (GH 1–171) that are classified based on amino acid sequences and structural similarities, which are all accessible in the constantly updated CAZy database at http://www.cazy.org, accessed on 10 March 2021 [16].

GHs enzymes can break down carbohydrates and polysaccharides; however, not all pathogen genomes contain genes coding for all the GH families, and some families are present in fungi not necessarily known for their ability to degrade cellulose [10]. So far, most identified and characterized GHs are from bacteria, in which their distribution across genomes is conserved within genera [17,18]. Many GH families with different activities have been detected in various fungal species. Although Ascomycetes and Basidiomycetes differ in the abundance of GH families, most glycoside hydrolases characterized from fungi have been reported to belong to the division Ascomycota [10,13]. Murphy and colleagues identified 453 fungal glycoside hydrolases from 131 fungal species which represented 49 different GH activities, and all were extracellular enzymes [13]. They indicated that the genus, *Aspergillus*, had the largest number of GH encoding genes. Comparative analysis of GH families across the entire fungal kingdom showed that Ascomycetes had more members of families GH2, GH72, and GH76 compared to Basidiomycetes. However, GH5, GH13, GH16, GH18 GH31, and GH61 are the most prevalent families among all fungi investigated to date [10]. In addition, it was reported that some GH families (e.g., GH7) mainly exist in fungi and oomycetes, but were not found in bacteria or archaea [19]. Based on the CAZy database, many characterized GHs show substrate specificity. For example, most identified proteins with domains from GH5, GH6, GH7, GH8, GH9, and GH12 families can act on cellulose, while GH18 and GH19 families act on chitin. However, some GH families show broad and mixed substrate specificity [18].

In recent years, the roles of CWDEs belonging to several GH families have been proved in invasion, pathogenicity or virulence of fungi and oomycetes on the plant host, but the exact mechanisms are still poorly understood [20,21,22,23,24]. Several studies indicated that pectinases [25], xylanases [26,27], xyloglucanases (XEGs) [28], cellulases [29,30,31], and chitinases [32], were related to pathogenicity of fungi. Pectate lyases were the first CWDE that were shown to be required for the full pathogenicity of *Erwinia chrysanthemi* causing the bacterial stem and root rot disease [33]. Since then, the role of many CWDEs in the virulence of various fungal pathogens has also been reported [28,30,31,34] (Figure 2). The role of pectate lyases as a virulence factor in the pathogenic fungi, *Colletotrichum coccodes* (a pathogen of potato and tomato), and *Verticillium dahliae* (a soil-borne fungus), has been well-described [35,36]. Moreover, disruption of both pectate lyase-genes in *Fusarium solani f. sp. pisi*, led to reduction of pathogenicity on pea [37]. Several studies have reported that some CWDEs, such as xyloglucanase (XEG1) [20,21,22,23,24] and endopolygalacturonases [38], can act as pathogen-associated molecular patterns (PAMPs), and induce PAMP-triggered immunity (PTI), in which their PAMP activity could be independent of their enzymatic activity. In *V. dahliae*, for example, a pectate lyase and a cutinase (VdPEL1 and VdCUT11, respectively) contributed to the fungal virulence and simultaneously activated plant immunity as PAMPs [36,39]. It has been reported that cellulases, xylanases, and cutinases play crucial roles in the virulence of *Magnaporthe oryzae*, the causal agent of rice blast disease [30,40]. CWDEs are important virulence factors in *F. graminearum*, the pathogen causing Fusarium head blight disease on wheat [41]. In addition, some CWDEs such as xylanases and hemicellulases are required for full pathogenicity of *Valsa mali* which causes a destructive disease on apple trees [42,43] (Table 1).

## 3. Chitinases (Glycoside Hydrolases Family 18 and 19)

GH families 18 and 19 in the CAZYme dataset include chitinolytic enzymes which are widely expressed in various organisms not only in fungi and insects, which contain chitin to their structures, but also to those who lack chitin in their cell walls such as plants, bacteria, and oomycetes [61,62]. GH18 family chitinases are predominant among fungi and plants. Chitinases from fungi are diverse and mainly reported in the GH family 18 which are associated with fungal development and autolysis [63,64]. Fungal chitinases were previously divided into three major clades known as A, B, and C; however, new phylogenetic analyses of GH18 proteins across many fungal genomes showed that chitinases previously assigned to the clade “C” were not distinct from the clade “A”, thus they were placed in one clade as AC. In addition, chitin-binding domains (CBDs) were not placed in certain specific clades and were identified widely across clades of chitinases [65].

Chitinases are responsible for the degradation of the β-(1,4) linkages of chitin, a major polysaccharide component of the fungal cell wall, leading to the release of *N*-acetylglucosamine oligomers [66,67]. Fungi deploy chitinases for various purposes. So far, the role of chitinases in the growth, nutrition, mycoparasitism, and virulence of fungi have been identified [55,63,68]. Filamentous fungi often contain expanded chitinase genes; for example, *A. nidulans* and *A. fumigatus* have 19 and 20 genes encoding GH18 enzymes, respectively. About 36 chitinase genes have been identified in *Trichoderma virens*, a mycoparasitic species [69]. The range of redundancy within the GH18 family can be problematic for studying the role of specific chitinase in fungi since it needs multiple deletion mutants for this purpose [70]. The genome of the fungus, *M. oryzae*, encodes 15 chitinases belonging to the GH18 family that are mostly essential for colonization of the host, suppression of host immunity, and fungal virulence [56].

Chitinases are also involved in fungal–fungal interactions. It has been shown that deletion of certain chitinase genes in *Clonostachys rosea* and *A. nidulans* results in a reduction of inhibitory activity against *Botrytis cinerea* and *Rhizoctonia solani* (Table 1) [57,58]. In plants, chitinases are classified into five classes and among them, classes III and V belong to the GH18 family, while Classes I, II, and IV belong to the GH19 family. Chitinases of the GH19 family have been reported mainly in plants and bacteria, and are thought to be produced as a defense mechanism against fungal invasion; thus, they are seldom found in fungi [62]. In this case, chitinases produced by plants can hydrolyze the chitin, the major component of the fungal cell wall. Released chitin oligomers act as PAMPs, which will be recognized by special chitin receptors including the chitin elicitor-binding protein (CEBiP) and the chitin elicitor receptor kinase (CERK1) [59] located in the plant cell surface and elicit a plant-defense response. Pathogenic fungi have developed different strategies to suppress chitin-triggered immunity in host plants, but the diversity of mechanisms that fungi use for suppression of plant response and pathogenicity is not clear [71]. For example, *Cladosporium fulvum*, the fungal pathogen of tomato, secretes two effector proteins known as extracellular protein 6 (ECP6) and avirulence protein 4 (AVR4), which bind to chitin molecules in the fungal cell wall to shield and protect the fungus against recognition by plants. AVR4 utilizes family 14 carbohydrate-binding module (CBM14) and ECP6 uses Lysin motif (LysM) domains to sequester chitin oligosaccharides released from hydrolysis of the fungal cell wall by plant chitinases to suppress elicitation of host immunity [72], [73]. Most chitin-binding effectors in fungi contain LysM domains which are conserved and broadly distributed in the genomes indicating their essential roles in binding and protecting chitin in fungal cell walls against chitinases secreted by a plant during infection [73,74]. These effectors, with similar functions, have also been identified in other plant fungal pathogens, such as *M. oryzae*, *Colletotrichum higginsianum*, and *R. solani* [59,75]. It was reported that in *M. oryzae*, LysM-containing effector protein (Slp1) competes with CEBiP for chitin binding, thereby preventing the activation of immune response in rice [59].

Two fungal species in Basidiomycete class, threatening cacao production in America, *Moniliophthora perniciosa* and *Mo. roreri* (causing witches’ broom disease and frosty pod rot on cacao, respectively), encode chitinase genes in the GH18 family known as *MpChi* and *MrChi*, respectively. These genes were reported to be highly expressed during the interaction with cacao plants and encode catalytically inactive chitinases, which despite the lack of enzymatic activity, still bind to chitin oligomers and prevent elicitation of chitin-triggered immunity [71]. In other words, these chitinases act as virulence factors that chelate free chitin, thus preventing the activation of the plant immune system. Other plant-pathogenic fungi (e.g., *Blumeria graminis*, *Colletotrichum* spp., *Fusarium* spp., *Puccinia* spp., *R. solani*, and *Va. mali*) seem to encode putative inactive GH18 chitinases suggesting that similar strategies are likely used by other fungal phytopathogens (Table 1) [71].

Soil-borne fungal pathogens secrete proteins that can alter chitin in the fungal cell wall. It has been shown that *V. dahliae* and *Fusarium* sp., among the most destructive soil-borne fungal species, secrete polysaccharide deacetylase (PDA1) that are essential for fungal virulence. This chitin deacetylase can modify chitin oligomers to chitosan, preventing the perception of chitin by plant chitin receptors, therefore, suppressing chitin-triggered host immunity [76].

Recently, Martínez-Cruz et al. [32] characterized a new family of effectors in fungal pathogens acting as chitinases that are called effectors with chitinase activity (EWCAs). These fungal effectors degrade chitin oligomers but they do not have chitinolytic activity against the fungal cell wall. They investigated the function of these EWCA-like proteins in the cucurbit powdery mildew which causes by the fungus *Podosphaera xanthii* [77]. They indicated that these fungal chitinases are important for suppression of chitin-triggered immunity by breaking down chitin oligomers into smaller molecules that cannot be recognized by plant chitin receptors, avoiding the perception of chitin by plant immune system, and suppressing chitin-triggered host immunity. These EWCA proteins are conserved in the genomes of various fungal pathogens indicating their crucial roles in the successful infection and pathogenicity of fungi (Table 1) [32].

## 4. Endoglucanases (Glycoside Hydrolase Families 5, 6, 7 and 45)

Cellulose is one of the major polysaccharides in the cell wall of most plants and is the substrate of cellulase enzymes. Cellulases are produced by various microorganisms such as fungi, oomycetes, bacteria, and actinomycetes and hydrolyze the β-1,4-glycosidic bonds in cellulose polymer. Cellulases compose a group of enzymes including endoglucases, exoglucanases, and ß-glucosidases [24,78]. Exoglucanases and endoglucanases act synergistically to produce cellobiose and are then cleaved by β-glucosidase to monosaccharides [38]. β-glucosidases are important in industrial biotechnology which hydrolyze glycosidic linkages and are distributed in glycoside families 1 and 3. Some cellulolytic fungi such as *Trichoderma* species are well recognized for their cellulose-degrading abilities and are typically used for biofuel production [79]. Several endoglucanases, exoglucanases, and *β*-glucosidases were isolated and purified from *T. viride* and *T. reesei* [80]. Although many fungi produce cellulases, the roles of these enzymes in pathogenicity and plant–pathogen interactions are not fully understood. This may be due to the multiplicity of cellulase genes with similar functions throughout the fungal genomes [81]. For example, the genome of the fungus, *M. oryzae*, contains three and six genes encoding GH6 and GH7 cellulases, respectively.

Enzymes belonging to GH7 cleave the β-1,4-glucosidic bonds in the cellulose chain [82]. Some GH7 members also act on xylan. GH7 enzymes are prevalent in fungi but have not been identified in bacteria or archaea [31]. It has been indicated that the GH6 and GH7 cellulases play important roles in the virulence of *M. oryzae*, involved in both penetration and expansion of the fungus in the host [30]. Construction of GH6 and GH7 multiple knockdown strains in *M. oryzae*, led to fewer lesions, less penetration, and infection of fewer cells. A high rate of papilla formation blocked the invasion of the knockdown mutants into host cells, indicating that GH6 and GH7 were involved in *M. oryzae* virulence. Interestingly, studies expressing CH7 cellulases of commercial value in heterologous systems have shown the importance of post-translational modifications (i.e., *N*-terminal glutamine cyclization) for proper enzymatic function (Table 1) [83].

The endo-1,4-β-glucanase encoding gene *AaK1,* which belongs to the GH6 family, was reported in phytopathogenic fungus *Alternaria alternata*, and supposed to be an important virulence factor, facilitating disease development and penetration in fruits, while it was related to an increasing environmental pH. Rising pH to 6.0 or higher, increased the expression of *AaK1* and production of endo-1,4-β-glucanase, which resulted in maximum virulence of *A. alternata* and its decay development on fruits [45]. Furthermore, a β-1,4- endoglucanase encoding gene (*egl1*) has been reported in the soil-borne deuteromycete fungus, *Macrophomina phaseolina* which was unique and differed from any known endoglucanases in fungal saprophytes, belonging to the GH5 family (Table 1). Since the substrate of egl1 was similar to a plant-encoded endoglucanase which was active during plant cell wall expansion, it was speculated that *egl1* was specific in phytopathogens and mimic the function of plant endoglucanases, facilitating surreptitious penetration of the cell wall [44].

In oomycetes, there is still little information available on the role of CWDEs. In *Phytophthora sojae*, a soil-borne oomycete causing soybean stem and root rot disease, the gene *PsGH7a* encoding a GH7 family cellulase was identified to be highly induced during infection, suggesting its important role in the invasion and virulence of this pathogen. In addition, this gene (*PsGH7a*) is highly conserved in both fungi and oomycetes and has a high percentage of identity for an amino acid sequence within the genus *Phytophthora* (Table 1) [31].

Cellulases can also act as elicitors in plant-pathogen interactions. The endoglucanase EG1, which belongs to the glycosyl hydrolase family 45, acts as an elicitor in the soil-borne plant pathogenic fungus, *R. solani*, which can induce cell death during infection. Moreover, it has been shown that full loss of its catalytic activity would not affect fungal infection suggesting that endoglucanase is an elicitor in fungi, but its cellulase activity is independent of its elicitor activity (Table 1) [60].

## 5. Xylanases (Glycoside Hydrolase Families 10 and 11)

Xylanases hydrolyze the β-1,4 bond in the xylan backbone of hemicellulose. This group of enzymes can be found in different glycoside hydrolase families, including GH5, GH7, GH8, GH10, GH11, and GH43 and have been reported in fungi, bacteria, and protozoa [84]. Most xylanases, which have been reported from fungi so far, belong to GH10 and GH11 families [85]. It has been reported that xylanases are required for the virulence of some phytopathogenic fungi. GH10 and GH11 endoxylanases were important pathogenicity factors in the fungus, *M. oryzae* during the infection process on rice. Since the reduction of virulence was higher in GH10 and GH11 mutants compared to GH6 and GH7 ones, it was speculated that GH10 and GH11 might have more important roles in cell wall penetration than the GH6 and GH7 cellulases (Table 1) [46]. Two genes (*MGG_14243.6* and *MGG_07868.6*) from *M. oryzae*, encoding for GH10 enzymes were up-regulated upon infection of host plants, suggesting involvement in fungal virulence [30]. Silencing of multiple GH10 genes in the same fungal species showed a significant decrease in xylanase activity and reduced infection lesions on barley plants, indicating a crucial role of these enzymes in pathogenicity (Table 1) [30].

The xylanase *xyn11A,* belonging to the GH11 family, was shown to be essential for full virulence of the necrotrophic fungus *B. cinerea*. Catalytically impaired Xyn11A mutants did not eliminate the ability of necrosis induced by fungus, showing that Xyn11A contributed to virulence independently of enzymatic activity. Therefore, it was reported that Xyn11A promoted the necrosis of the plant tissue surrounding the infected plant areas, allowing the fungus to grow rapidly on necrotic tissues (Table 1) [47,48]. The same results had been previously observed in xylanases from other fungi. For example, it was shown that xylanases from *Trichoderma* species, contributed to virulence by eliciting necrosis in plant leaves, but its enzymatic activity was not required for its elicitor activity [49].

A protein from the GH11 family called Vd424Y has been recently reported in the soil-borne fungus, *V. dahliae*, and is an important effector protein targeting the plant nucleus and is necessary for full virulence of *V. dahliae* on its host (Table 1). It has been shown that both signal peptide and the nuclear localization signal are essential for Vd424Y-induced cell death. The transient expression of Vd424Y in *Nicotiana benthamiana* also induced plant cell death mediated by leucine-rich repeat (LRR) receptor-like kinases (PRR-RLKs), BAK1, and SOBIR1, indicating that Vd424Y is a potential PAMP that is able to activate PTI responses in plant [50].

## 6. Xyloglucanases (Glycoside Hydrolase Families 12 and 74)

Xyloglucanases (XEG) are one of the CWDEs that specifically break down xyloglucan, a major hemicellulosic component of the plant cell wall consisting of β-1,4-glucan linkages. They are found in bacteria, plants, and fungi and they mostly belong to GH12, GH74, and GH5 in the CAZY database [28,86,87]. To date, most of the xyloglucanases that have been studied in fungi belong to family GH12 and GH74 [20,28]. The structures of XEG enzymes in *Clostridium thermocellum* and Aspergillus niger have been well characterized [88,89]. It was reported that XEG belonging to the GH74 family acts as a virulence factor, which plays a significant role in the pathogenicity of *Coniella vitis*, the causal agent of grape white rot disease in China. The results of a recent study indicated the high level of hydrolytic activities of both xylanase and XEG in *C*. *vitis* during infection, while the deletion of XEG-coding gene *CvGH74A* decreased disease development suggesting a key role of XEG in the pathogenicity of the fungus (Table 1) [28].

Sato and colleagues reported that the hydrolytic activity of XEG was very high in the soil-borne fungus *V. dahliae*, but its role in pathogenicity was not clear [86]. However, Gui et al., [20] indicated that two of the six GH12 proteins produced by *V. dahliae* (VdEG1 and VdEG3) are virulence factors that act as PAPMs to trigger PTI regardless of their enzymatic activity (Table 1).

Recently, five proteins with GH12 domains have been found in the soil-borne vascular phytopathogenic fungus, *F. oxysporum*, among which, FoEG1 plays a crucial role in fungal pathogenicity as a virulence factor during interaction with its host [51]. FoEG1 is an apoplastic cell death-inducing protein, which is similar to VdEG1 from *V. dahliae* [20] and could be a PAMP to trigger plant cell death mediated by PRR-RLKs, BAK1, and SOBIR1 (Table 1). Thus, the results show that FoEG1 in *F. oxysporum* is an important virulence factor during infection and colonization by the fungus [51].

Genome mining of different oomycete, fungal, bacterial, and plant species showed that GH12 homologous-protein sequences of XEG1 were present and common in microbial taxa, but not in plants. In addition, among oomycete plant pathogens, a considerable amount of GH12 proteins were identified in *Phytophthora* species and in *Hyaloperonospora arabidopsidis*, but not in the necrotrophic oomycete, *Pythium ultimum* [24]. In *P. sojae*, the causal agent of stem and root rot of soybean, it was shown that XEG1 with xyloglucanase and β-glucanase activity could act as an important virulence factor during pathogen infection. In addition, XEG1 was recognized as a PAPMP and triggered an immune response in soybean and solanaceous species. Moreover, although XGE1 could be recognized as a PAPM by the host plant, several RXLR effectors secreted by *P. sojae* are able to suppress the plant-defense response triggered by XEG1, which underscores XGE1′s role in pathogenicity (Table 1) [24].

## 7. α-Amylases (Glycoside Hydrolase Family 13)

There is mounting evidence that α-amylases, which belong to the GH13 family, maybe involved in stealth pathogenicity strategies by fungi to avoid host detection. The poplar pathogens, *Mycosphaerella populorum* and *My. populicola*, have a cluster composed of an α-glucan synthase and α-amylase that are co-expressed only when grown in the presence of wood chips [52]. Dhillon and colleagues went on to suggest that these fungi may be masking the chitin in their cell walls with α-glucan to avoid host detection [52]. This cluster seems to be present in other fungal genomes, including some that are human pathogens. Similar strategies have been attributed to *M. oryzae*, *R. solani*, and *Cochliobolus miyabeanus*, since these fungi also mask their cell walls with α-1,3-glucan, which cannot be degraded by the plant. The α-1,3-glucan seems to offer protection against antifungal agents produced by the plant, limiting the production of PAMPs and thereby reducing the possibility of PTI [53]. Stealth pathogenicity conferred by α-amylases has also been attributed to the wheat pathogen, *Mycosphaerella graminicola* (Table 1) [52,54].

## 8. Conclusions

Despite progress in understanding the various CWDEs with different glycoside hydrolase protein domains, there is still little knowledge on the role of these enzymes in the virulence of fungi. This is due to the extensive redundancy of genes encoding CWDEs in the fungal genome. Based on the genome sequence of plant pathogenic fungi, there are often more than 10 or 20 genes in a group of CWDEs in the genome. However, the results from many studies support the involvement and importance of several GH protein families in the virulence of phytopathogenic fungi. While major attention has been given to plant pathogenic fungi, the roles of CWDEs in oomycetes are still poorly understood, suggesting further investigation is needed to assess the roles of these enzymes in fungal-host interactions.

## Figures and Tables

**Figure 2 ijms-22-09359-f002:**
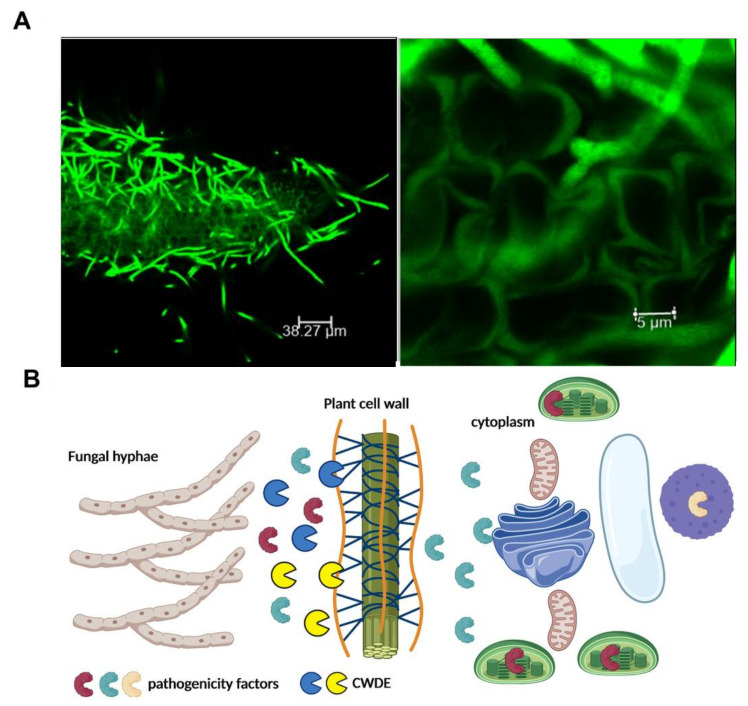
CWDEs are involved in host plant colonization. (**A**) Confocal scanning microscopy of *Arabidopsis thaliana* roots infected with the soil-borne pathogen, *Verticillium longisporum*, expressing the green fluorescent protein. To avoid recognition and successfully colonize its host, *V. longisporum* produces small secreted proteins as well as a repertoire of carbohydrate-active enzymes to degrade plant material [34]. Photo by Dr. Panagiotis Moschou (https://pmoschoulab.wordpress.com/home/, accessed on 10 March 2021). Used with permission. (**B**) Schematic representation of the role of CWDEs in the establishment of a successful infection. Fungi secrete CDWEs in order to facilitate the penetration of pathogenicity factors to the plant cytoplasm. The figure was created by BioRender software.

**Table 1 ijms-22-09359-t001:** Known glycoside hydrolase families in phytopathogenic fungi and oomycetes and their potential function.

Family	Enzymatic Activity	Fungal/Oomycete Species	Role	References
GH5	Endoglucanases; cellulasese	*M. phaseolina*	Penetration of the cell wall	[44]
GH6	Endoglucanases; cellulases	*M. oryzae*; *A. alternata*	Penetration and expansion of the fungus in the host	[30,45]
GH7	Endo-β-1,4-glucanases; cellulases	*M. oryzae*; *P. sojae*	Virulence factor	[30,31]
GH10	Endo-1,4-β-xylanases	*M. oryzae*	Virulence factor	[46]
GH11	Xylanases	*M. oryzae*; *B. cinerea*; *Trichoderma* sp.; *V. dahliae*	Virulence factor	[47,48,49,50]
GH12	Xyloglucanases	*F. oxysporum*; *V. dahliae*; *P. sojae*	Virulence factor; PAPM activity and inducing plant immune response	[20,24,51]
GH13	α-amylases	*M. populorum*; *M. populicola*; *M. oryzae*; *R. solani*; *C. miyabeanus*; *M. graminicola*	Stealth pathogenicity	[52,53,54]
GH18	Chitinases	*T. viride*; *C. rosea*; *A. nidulans*; *M. oryzae*; *Mo. perniciosa*; *Mo. roreri*; *P. xanthii*	Growth, nutrition, mycoparasitism, virulence factor; suppression of chitin-triggered immunity	[32,55,56,57,58,59]
GH45	Endocellulases; Endoglucanases	*R. solani*	PAMP activity and inducing plant immune response (PTI)	[60]
GH74	Xyloglucanases	*C. vitis*	Key virulence factor	[28]

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
