# Peer review of "The Role of Glycoside Hydrolases in Phytopathogenic Fungi and Oomycetes Virulence"

_ijms, 2021, doi:10.3390/ijms22179359_

Round 1

Reviewer 1 Report

It is difficult to follow the descriptions because no considerations for the correlations between text body and "Figures and Table".

 1) In text body :"cellulose, hemicellulose and pectin (Fig. 1) [3]. To 27" But no description for "pectin" in Fig 1.

2) In the text body:"Since then, the role of many CWDEs in the 99 virulence of various fungal pathogens has also been reported [28], [30], [31], [34] (Fig. 2). 100" 

In Figure2 "Verticillium longisporum, expressing the green fluorescent protein." can not recognize that "CWDE" was modified by green fluorescent or not.

Readers can recognize that Verticillium longisporum colonize its host, but can not recognize that "CWDE" played critical roles for colonization in this Figure.

3) It is very difficult understand In Table 1.

First there is no description why did the authors summarize Table 1 without no reference.

Orders the families in the table will be identical with text body descriptions. (GH19 will be added in the table.)

 Is it different  between "Xyloglucan hydrolases" in GH12 and  "Xyloglucanases" in GH74.

4) Each description for families after section 4. For example

It is not easy to recognize description "4. Glycoside hydrolase> families 6, 7 and 45" .

Section title will be change based on "Endoglucanasee (Families 6, 7 and 45)"

5) There is no detail descriptions for GH 6. For example, "the ge- 218 nome of the fungus, M. oryzae, contains three and six genes encoding GH6 and GH7 cel- 219 lulases, respectively. 220 Enzymes belong to GH7 cleave the β-1,4-glucosidic bonds in cellulose chain [70].221 

  Some GH7 members also act on xylan. GH7 enzymes are prevalent in fungi, but have not identified in bacterial and archaea. It has been indicated that the GH6 and GH7 223 cellulases play important roles in the virulence of M. oryzae, involved in both penetration 224 and expansion of the fungus in the host [30]."   These sentences are description of the role GH7. Many readers cannot recognize the GH 6 plays important roles  from these sentences.    6) There is no detail descriptions for GH 10 same as described above.   Authors have to reconstruct the review with consideration to read easy for many readers. 

Author Response

It is difficult to follow the descriptions because no considerations for the correlations between text body and "Figures and Table".

Answer: We would like to thank the reviewer for the constructive comments. We tried to address to address all comments and we hope that the revised version of our manuscript would be considered suitable for publication.

 1) In text body :"cellulose, hemicellulose and pectin (Fig. 1) [3]. To 27" But no description for "pectin" in Fig 1.

Answer:  The structure of pectin has been added to figure 1. New figure has been added to the revised version of the manuscript.

2) In the text body:"Since then, the role of many CWDEs in the virulence of various fungal pathogens has also been reported [28], [30], [31], [34] (Fig. 2). In Figure2 "Verticillium longisporum, expressing the green fluorescent protein." can not recognize that "CWDE" was modified by green fluorescent or not. Readers can recognize that Verticillium longisporum colonize its host, but can not recognize that "CWDE" played critical roles for colonization in this Figure.

Answer:  We have revised figure 2 and we added a schematic representation in order to show the role of CWDEs to host colonization and establishment of infection.

3) It is very difficult understand In Table 1.

First there is no description why did the authors summarize Table 1 without no reference.Orders the families in the table will be identical with text body descriptions. (GH19 will be added in the table.). Is it different  between "Xyloglucan hydrolases" in GH12 and  "Xyloglucanases" in GH74.

Answer: We have chosen to present a summary of this review in a table and to keep them in numerical order to make it easier for the readers to follow it. We also added references in a separate column. Since GH19 chitinases have not been reported to fungi, we did not add to the table. According to CAZy database Xyloglucan hydrolases and Xyloglucanases are the same (EC 3.2.1.151). Thus, the term Xyloglucanases is used in the revised manuscript.

4) Each description for families after section 4. For example

It is not easy to recognize description "4. Glycoside hydrolase families 6, 7 and 45".Section title will be change based on "Endoglucanasee (Families 6, 7 and 45)".

Answer: We have changed the titles accordingly: ‘’ Chitinases (Glycoside hydrolases family 18 and 19), Xylanases (Glycoside hydrolase families 10 and 11), Xyloglucanases (Glycoside hydrolase families 12 and 74), α-amylases (Glycoside hydrolase family 13).     

5) There is no detail descriptions for GH 6. For example, "the genome of the fungus, M. oryzae, contains three and six genes encoding GH6 and GH7 cellulases, respectively. Enzymes belong to GH7 cleave the β-1,4-glucosidic bonds in cellulose chain [70]. 

Some GH7 members also act on xylan. GH7 enzymes are prevalent in fungi, but have not identified in bacterial and archaea. It has been indicated that the GH6 and GH7 cellulases play important roles in the virulence of M. oryzae, involved in both penetration  and expansion of the fungus in the host [30]. These sentences are description of the role GH7. Many readers cannot recognize the GH 6 plays important roles  from these sentences.   

Answer: in this sentence the contribution of both GH6 and GH7 in virulence has been described. We have also expanded the description of GH6 and GH7: ‘’ Construction of GH6 and GH7 multiple knock down strains in M. oryzae, led to fewer lesions, less penetration and infection of fewer cells. A high rate of papilla formation blocked invasion of the knock down mutants into host cells, indicating that GH6 and GH7 involved in M. oryzae virulence’’.

We also added a description about the AaK1 in A. alternata in the following paragraph to make clear that it also belongs to GH6 family.

 6) There is no detail descriptions for GH 10 same as described above.  Authors have to reconstruct the review with consideration to read easy for many readers. 

Answer: A detailed description of GH10 has been added to the revised version: ‘’ Two genes (MGG_14243.6 and MGG_07868.6) from M. oryzae, encoding for GH10 enzymes were up-regulated upon infection of host plants, suggesting involvement in fungal vir-ulence [30]. Silencing of multiple GH10 genes in the same fungal species, showed a significant decrease in xylanase activity and reduced infection lesions on barley plants, indicating a crucial role of these enzymes in pathogenicity [30]’’.

Reviewer 2 Report

The manuscript entitled" The role of glycoside hydrolases in phytopathogenic fungi and oomycetes virulence is a well-written review article presented by the authors.

The information is highly relevant and well presented.

  1. However, I suggest if authors can present some schematic or graphical figures to represent the role and mechanism will be highly relevant and will be interesting for readers to well understand—
  2. Glycoside hydrolase> families 6, 7, and 45 – what does this > symbol means- please indicate.
  3. Figure 2- I don’t think figure 2 is required. If authors can make some pictorial scheme will be a better presentation.

Author Response

We would like to thank the reviewer for the constructive comments. We have now tried to address all points and we hope that the revised version of our manuscript would be considered suitable for publication.

The manuscript entitled" The role of glycoside hydrolases in phytopathogenic fungi and oomycetes virulence is a well-written review article presented by the authors.

The information is highly relevant and well presented.

  1. However, I suggest if authors can present some schematic or graphical figures to represent the role and mechanism will be highly relevant and will be interesting for readers to well understand.

Answer: We have modified figure 2 in order to include a schematic representation of the role of CWDEs in fungal colonization.   

  1. Glycoside hydrolase> families 6, 7, and 45 – what does this > symbol means- please indicate.

Answer: the symbol > is wrong and has been deleted.

  1. Figure 2- I don’t think figure 2 is required. If authors can make some pictorial scheme will be a better presentation.

Answer: The figure 2 has been modified in order to be more informative including a pictorial scheme as well.

Round 2

Reviewer 1 Report

Revised manuscript will be acceptale for the journal.